# Effect of Transcranial Direct Current Stimulation on Endurance Performance in Elite Female Rowers: A Pilot, Single-Blinded Study

**DOI:** 10.3390/brainsci12050541

**Published:** 2022-04-23

**Authors:** Zhiqiang Liang, Junhong Zhou, Fujia Jiao, Trenton Gin, Xi Wang, Yu Liu, Jiaojiao Lü

**Affiliations:** 1School of Kinesiology, Shanghai University of Sport, Shanghai 200438, China; 1711042011@nbu.edu.cn (Z.L.); jiaofujia@gmail.com (F.J.); 1811516022@sus.edu.cn (X.W.); lvjiaojiao@sus.edu.cn (J.L.); 2Key Laboratory of Exercise and Health Sciences of Ministry of Education, Shanghai University of Sport, Shanghai 200438, China; 3Harvard Medical School, Harvard University, Boston, MA 02131, USA; junhongzhou@hsl.harvard.edu; 4The Hinda and Arthur Marcus Institute for Aging Research, Hebrew Senior Life, Roslindale, MA 02131, USA; 5College of Art and Science, Cornell University, New York, NY 14850, USA; trentongin2021@gmail.com

**Keywords:** rowing, endurance performance, tDCS, ergogenic effect, elite athlete

## Abstract

Endurance, which is dependent at least partly upon the activation of the brain cortex, is important for performance in rowing. Transcranial direct current stimulation (tDCS) has shown benefits for endurance, but its effects on the endurance performance of elite rowing athletes are unknown, and are examined in this study. Eight elite female rowers completed 5 km of rowing on an ergometer following stimulation of the tDCS and sham over motor cortices. Each session lasted 20 min and the current was set at 2.2 mA. Time, 500 m/split, power, time corresponding to 500 m (TC500) and power corresponding to 500 m (PC500) were recorded continuously throughout the tests. No significant differences in time, 500 m/split and power were observed between baseline, tDCS and sham. Compared to the sham, tDCS induced a percentage reduction in TC500 from baseline to 2500 m and 4000 m, and a percentage increase in PC500 from baseline to 500 m, 1000 m, 1500 m, 2000 m, 2500 m, 4000 m, 4500 m and 5000 m. One-session tDCS did not have significant benefits for rowing endurance performance in elite professional rowers, and had only marginally greater efficacy compared to sham. These findings offer knowledge helpful to the design of future studies exploring the effects of tDCS on the endurance performance of elite rowers.

## 1. Introduction

Endurance capacity, which depends upon aerobic metabolism, is critical to performance in rowing [1]. Traditional training for rowers is usually of long duration and low intensity, with the goal of improving endurance capacity, but it often induces fatigue, affecting the endurance of rowers [2,3]. Endurance capacity is a relatively complex basic ability; it consists of the peripheral elements and multiple central elements [4]. Recent studies have observed that central elements, including the cortical regions, of the brain play an important regulatory role in endurance performance [5], and have linked greater cerebral excitability in brain regions pertaining to cognitive–motor control to an increased output of neural drive and spinal motor neuron excitability, effectively alleviating fatigue and increasing endurance during exercise [6,7,8]. Therefore, strategies designed to facilitate the excitability of cortical regions within the brain hold great promise in enhancing endurance capacity, which will ultimately help to improve rowing performance.

tDCS is one such technique that can non-invasively and safely modulate cortical activities by sending direct micro-level currents to the targeted regions via scalp electrodes [9,10]. In recent decades, more attention has been paid to the ergogenic effects of tDCS on endurance performance [5,11]. Vieira et al. [12] and Fores et al. [13], for example, showed that using transcranial direct current stimulation (tDCS) to modulate the cortical excitability of the prefrontal cortex can significantly improve the endurance performance of athletes in back squat and swimming. In another study, Wang et al. [8] reported that bilaterally modulating the cortical excitability of the primary motor cortex (M1) using tDCS can significantly improve muscle endurance performance in elbow flexion tasks.

By using the Halo Sport system of tDCS, Park et al. [14] showed that one session of tDCS targeting M1 significantly improved endurance running performance at 80% VO2max in trained runners. This finding suggests that using tDCS to target M1 via the Halo system can also improve the endurance performance. Nevertheless, the effects of such a tDCS protocol on endurance in elite rowing athletes have not been well characterized.

Therefore, we have here completed a pilot, single-blinded study consisting of eight elite athletes from the national rowing team of China and tested the effects of one-session tDCS targeting M1 via the Halo system on endurance performance. We hypothesized that, compared to the sham, tDCS would induce significantly greater endurance performance during rowing (e.g., higher power and shorter time) in these elite athletes.

## 2. Materials and Methods

### 2.1. Subjects

Eight national female rowers (height: 179.5 ± 2.17 cm, weight: 73 ± 4.94 kg) volunteered for this study. The eligibility criteria were (1) sculling techniques, (2) competitive performance ranking in the top 8 in the national regatta and (3) the regular completion of daily training. Rowers were not included in this study if they had any lower-limb and/or lower-back injury or pain within six months before the study. Each rower and her coach were informed of the experimental procedure and invited to provide their informed consent to participate in this study. The study protocol was approved by the institutional review board committee of Shanghai University of Sport (102772019RT020).

### 2.2. Study Protocol

The rowers visited the training center three times, which included one baseline visit to measure endurance performance and two intervention visits. Participants were selected in a single-blinded, parallel, randomized, counterbalanced order to receive tDCS and sham stimulation. The rowers were asked to not consume coffee or alcohol and to sleep for at least 7 h before and during the test period. All visits were separated by at least 72 h, and each visit was arranged at the same time of the day.

During the first visit, the rowers were familiarized with the experimental procedures, and we set up the drag factor on the air damper of a rowing ergometer, which prevented any changes over the following 2–3 visits. In addition, the athletes executed a 5 km endurance test on a rowing ergometer to establish individual baseline performance. In this test, the rowers first performed a warm-up exercise for 15 min, followed by 5 km of rowing with a constant load at a 20 stroke/rate (SR). In visits 2–3, the rowers received the tDCS/sham stimulations via Halo Sport for 20 min after a warm up, and completed a constant-load 5 km row at 20 SR. During rowing, the rowers were allowed to receive verbal encouragement from performance coaches in order to achieve optimal performance.

### 2.3. Transcranial Direct Current Stimulation (tDCS) Procedure

The tDCS was administered by Halo Sport (© Halo Neuroscience, San Francisco, CA, USA) with three foam electrodes (size 4 × 6 cm) wetted with 0.9% NaCl that contained an electric current stimulator, driven by a lithium-ion (LiPo) cell (36 V). The tDCS montage was similar to that used by Huang et al. [15], which has been shown effective in whole-body exercises such as running [14] and ski jumping [16]. According to our stimulated results pertaining to the electric field, this montage could stimulate the motor cortex, the supplementary motor cortex and the primary and the secondary somatosensory cortices together. An anodal stimulator was placed over M1 (Cz, according to the 10–20 international EEG system), while the bilateral cathodal stimulators were placed over C5 and C6 (Figure 1). In the sham stimulation, the same setup was applied for anodal stimulation. The stimulation intensity was 2.2 mA and was controlled by the IOS software system on an iPhone or iPad. During the tDCS, the current was delivered for 20 min, whereas during the sham, the current was only applied for 30 s.

### 2.4. Endurance Training

At baseline and after the tDCS/sham stimulation, the rowers performed a constant-load 5 km row at 20 SR on a rowing ergometer (Concept Ⅱ, Morrisville, NC, USA). The 5 km row was preceded by a warm-up exercise lasting 15 min, as per the training plan given by the strength and conditioning coach. The rowers were verbally encouraged throughout the 5 km row by their performance coaches. Visual feedback of their performance was given by the PM5 display setting on the rowing ergometer (Concept Ⅱ, Morrisville, NC, USA). Time, 500 m/split, power, time corresponding to 500 m (TC500) and power corresponding to 500 m (PC500) were recorded, respectively, using MP5. All parameter data were exported using the Concept 2 utility (Concept, Morrisville, NC, USA).

### 2.5. Statistical Analysis

Unless specified, data are presented as mean ± SD. The normality of distribution was checked using the Shapiro–Wilk test, and the sphericity was checked using Mauchly’s test; the Greenhouse–Geiiser test was used to determine when the results violated the Mauchly’s test. A paired-sample t test was used to compare performance between two intervention visits. One-way ANOVA was used to compare the effects of tDCS on the time, 500 m/split and power between baseline, tDCS and sham conditions during 5 km of rowing. The post-hoc analysis of the Tukey test was used to check where there was statistical significance. The linear mixed-effects model was used to examine the effects of tDCS on TC500 and PC500 during the 5 km row. Statistical significance was set at *p* < 0.05. Statistical analysis was performed via SPSS 26.0 (IBM, New York, NY, USA).

## 3. Results

All participants completed all the tests. No side effects or injuries were reported. No significant difference was observed in endurance time between two intervention visits, during which the interventions were operated in a counterbalanced order (*t* = −0.421, *p* = 0.685).

### 3.1. Effect of tDCS on Rowing Endurance Performance

After tDCS, the performance of seven rowers improved as a result of tDCS, but only five were improved by the sham; the ANOVA models showed no significant differences in time (F = 0.251, *p* = 0.780), 500 m/split (F = 0.263, *p* = 0.772) or power (F = 0.283, *p* = 0.757) between baseline, tDCS and sham (Figure 2 and Table 1). The endurance performance of rowers was improved 1.05% from baseline after tDCS, but only 0.24% after sham.

### 3.2. Effect of tDCS on Time Corresponding to 500 m (TC500) and Power Corresponding to 500 m (PC500)

The linear mixed-effects models revealed significant differences in the percent changes in TC500 between tDCS and sham at 2500 m (*t* = −4.50, *p* = 0.011, ES = 0.91) and 4000 m (*t* = 3.11, *p* = 0.036, ES = 0.84); that is, after receiving tDCS, the TC500 was significantly shorter than that after sham (Figure 3 and Table 2). Significant differences in the percent changes in PC500 were also observed after tDCS, and in that after sham at 500 m (*t* = 65.67, *p* = 0.048, ES = 0.99), 1000 m (*t* = 22.47, *p* = 0.005, ES = 0.99), 1500 m (*t* = 26.22, *p* = 0.041, ES = 0.99), 2000 m (*t* = 20.06, *p* = 0.001, ES = 0.99), 2500 m (*t* = 19.01, *p* = 0.006, ES = 0.99), 4000 m (*t* = 11.11, *p* = 0.001, ES = 0.98), 4500 m (*t* = 21.03, *p* = 0.016, ES = 0.99) and 5000 m (*t* = 41.32, *p* = 0.001, ES = 0.99).

## 4. Discussion

This pilot study consisting of elite rowers from the national team suggested that tDCS targeting M1 induced no significant improvements in endurance performance during a 5 km row as compared to sham treatment. Only a marginally greater percent improvement was observed after tDCS. The major limitations are the small sample size and the single-blinded nature of the stimulation in this pilot study. However, the knowledge obtained from this pilot study may provide useful preliminary insights related to the design of protocols in future studies examining the effects of tDCS on the endurance of elite rowing athletes.

A single-blinded, randomized design was used in this study. It was observed that after sham treatment, the endurance performance was also improved, indicating a placebo effect. We used the inactive sham protocol due to the limitations of the device [15,17], which may induce a similar feeling to tDCS, but also potential physiological changes in the brain cortex [18]. Therefore, a novel active sham that can provide greater blinding efficacy without inducing changes in neuronal excitability is highly demanded [19,20]. For instance, Franceso et al. [19] reported that, compared to the inactive sham, the active sham with current flow navigated by the neuro-modeling technique could produce a similar scalp sensation to tDCS, by delivering constant lower-intensity currents, without affecting corticospinal excitability. Future studies implementing advanced neuro-modeling techniques are required to more explicitly examine the effects of tDCS on endurance performance.

We have here only examined the immediate effects of one 20 min session of tDCS. Studies have demonstrated that one session of tDCS can only improve functional performance for around 1.5 h [21]. Recent studies have further demonstrated that, compared to a single session of tDCS, multiple repeated sessions of tDCS over a period of days can induce cumulative effects (i.e., greater and longer-lasting) on performance, which can last for several weeks [21,22,23]. For instance, Pilloni et al. [24] reported that 10 sessions of 20 min tDCS over 10 days induced cumulative effects on walking and endurance performance in a multiple sclerosis population. Hilgenstock et al. [25] also reported that one session of 20 min tDCS over 5 days induced cumulative effects on motor learning in young individuals. Therefore, future studies may implement multiple sessions of tDCS intervention to determine whether such repeated sessions can augment the effects of tDCS on the endurance performance of elite rowers.

To maximize the effects of tDCS, novel techniques to personalize the montage of tDCS to each participant (including optimizing the cortical target placement of electrodes and current parameters) [26] are needed. Studies have demonstrated that an individualized design of tDCS montage can optimize the effects of tDCS due to the reversal of maladaptive processes and the enhancement of neuronal plasticity [27]. Studies have implemented several approaches to develop this design, such as computational modeling, Bayesian optimization and functional magnetic resonance imaging (fMRI). For instance, Da Silva et al. [28] reported that computer-modeling electric field simulation can determine the electrode position and guide the placement of tDCS for each participant; Datta et al. [29] reported that using models based on the brain anatomy derived from MRI data for each participant can normalize the inter-individual variation in current flow and electrical field delivered by the tDCS. It is thus highly demanded of future studies to implement a personalized montage of tDCS and examine its effects on endurance performance.

The small, single-gender sample size (athlete: female rowers, *n* = 8) is the most significant limitation of this study, as it induced high inter-subject variance and affected the statistical power, resulting in non-significant results. It is thus necessary to explore methods to measure the effects of tDCS in elite athletes. The second limitation of this study is the influence of the electrical field. We found that both lower- and upper-limb motor areas were stimulated by the montage used in Halo Sport, depending on the stimulation of the electrical field. However, we could not confirm which contributes more to rowing endurance performance. This should thus be confirmed in the future. The third limitation of this study is the lack of any statistical analysis of the placebo effect. Because it is important to estimate how placebo effects influenced the results, these should be analyzed in the future. The last limitation of this study is the lack of neurophysiological assessments performed to characterize the underlying neurophysiological changes induced by tDCS. Previous studies reported that endurance exercise requires an increase in the input into the spinal motor neurons in order to produce muscle force and power continually [30]. This is not solely dependent on the training of physical capacity, but can also be affected by the cortical excitability of the motor cortex [31]. Activating cortical excitability can benefit endurance performance [4] via enhancing synaptic connection [21], delaying the development of fatigue and decreasing the rating of perceived exertion (RPE) [12,32]. Therefore, examining the neurophysiological mechanisms of brain activity related to endurance using EEG and MRI techniques may help us to better understand the neurophysiology of the effects of tDCS on endurance performance.

## Figures and Tables

**Figure 1 brainsci-12-00541-f001:**
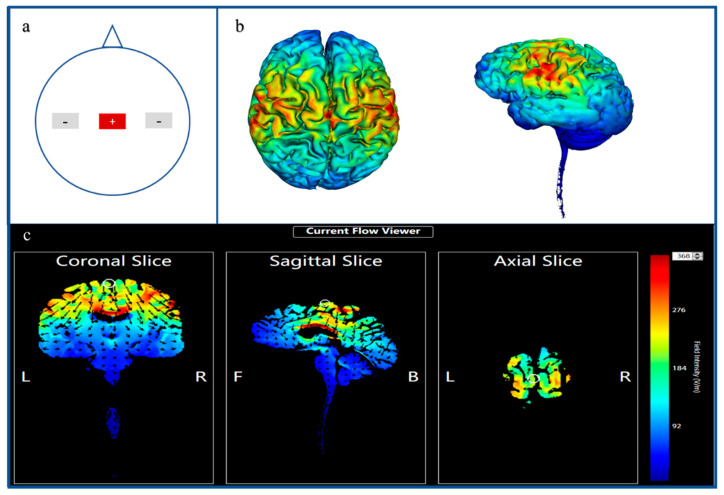
Overall review of transcranial direct current stimulation montages and simulation of electric field. (**a**) shows the location of the stimulated montage. The red rectangle shows the anodal stimulation area (Cz); the two gray rectangles depict the cathodal stimulation (left C5, right C6). (**b**) shows the hot point of the cerebral cortex caused by tDCS stimulation. (**c**) shows the distribution of the electric field of the tDCS at the coronal, sagittal and axial views, which were executed using the Target software.

**Figure 2 brainsci-12-00541-f002:**
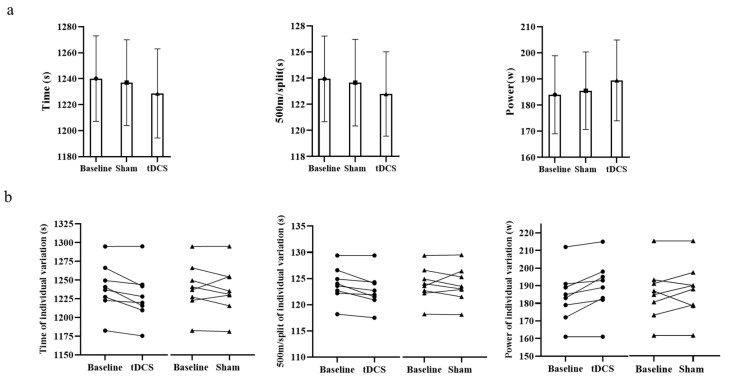
Overview of endurance performance and individual variation in time, 500 m/split and power during 5 km row. (**a**) is the endurance performance at baseline, transcranial direct current stimulation (DCS) and sham conditions. No significant differences were found between the three conditions as regards time, 500 m/split or power. (**b**) is the variation in individual endurance performance between tDCS and sham; seven of the rowers improved following tDCS, and five improved following sham. Data are presented as mean ± SD; *n* = 8.

**Figure 3 brainsci-12-00541-f003:**
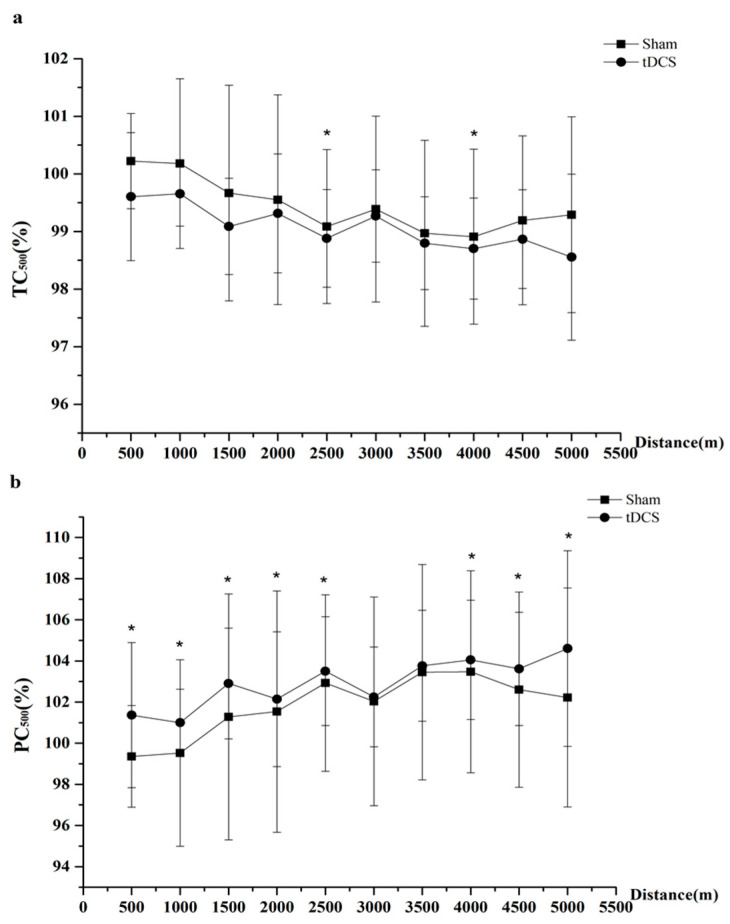
Effect of transcranial direct current stimulation (tDCS) on time corresponding to 500 m (TC500) and power corresponding to 500 m (PC500) during 5 km row. (**a**) is the percent changes in TC500 between tDCS and sham. Significant differences between tDCS and sham were found at 2500 m and 4000 m. (**b**) is the percent changes in PC500 between tDCS and sham. Significant differences between tDCS and sham were found at 500 m, 1000 m, 1500 m, 2000 m, 2500 m, 4000 m, 4500 m and 5000 m.

**Table 1 brainsci-12-00541-t001:** Overview of endurance performance of elite professional female rowers at baseline, tDCS and sham during 5 km row (data are presented as mean ± SD; *n* = 8).

Group	M ± SD	F	*df*	*p*	95% IC
Time (s)					
Baseline	1240.09 ± 32.91				1212.57–1267.60
Shame	1237.0 ± 533.03	0.251	(2,21)	0.780	1209.44–1264.67
tDCS	1228.65 ± 34.31				1199.97–1257.33
500 m/split (s)					
Baseline	123.95 ± 3.28				121.21–126.69
Shame	123.66 ± 3.31	0.263	(2,21)	0.772	121.89–126.42
tDCS	122.79 ± 3.23				119.93–125.64
Power (w)					
Baseline	184.00 ± 14.92				171.53–196.47
Shame	185.56 ± 14.84	0.283	(2,21)	0.757	173.16–197.97
tDCS	189.50 ± 15.47				176.56–202.44

**Table 2 brainsci-12-00541-t002:** Effect of transcranial direct current stimulation (tDCS) on time corresponding to 500 m (TC500) and power corresponding to 500 m (PC500) during 5 km row (data are presented as mean ± SD; *n* = 8).

Index	Baseline	tDCS	Sham	tDCS vs. Baseline (%)	Sham vs. Baseline (%)	*df*	*t*	*p*
TC_500_								
500 m	61.23 ± 1.68	60.98 ± 1.77	61.36 ± 1.56	99.60	100.22	4.00	−1.28	0.270
1000 m	61.97 ± 1.50	61.76 ± 1.61	62.08 ± 1.58	99.65	100.17	4.00	−0.85	0.445
1500 m	62.10 ± 1.68	61.53 ± 1.68	61.88 ± 1.59	99.08	99.66	4.00	0.12	0.912
2000 m	62.13 ± 1.63	61.71 ± 1.80	61.85 ± 1.67	99.31	99.55	4.00	1.55	0.197
2500 m	62.33 ± 1.69	61.63 ± 1.65	61.75 ± 1.66	98.88	99.08	4.00	−4.50	0.011 *
3000 m	62.26 ± 1.61	61.80 ± 1.67	61.87 ± 1.62	99.26	99.38	4.00	1.80	0.146
3500 m	62.29 ± 1.71	61.54 ± 1.74	61.64 ± 1.72	98.79	98.96	4.00	−1.86	0.136
4000 m	62.35 ± 1.81	61.53 ± 1.76	61.66 ± 1.74	98.7	98.91	4.00	3.11	0.036 *
4500 m	62.03 ± 1.76	61.32 ± 1.72	61.52 ± 1.78	98.86	99.19	4.00	0.19	0.859
5000 m	61.30 ± 1.87	60.41 ± 2.00	60.85 ± 1.83	98.55	99.28	4.00	−2.18	0.095
PC_500_								
500 m	191.43 ± 15.94	194.00 ± 16.53	190.06 ± 14.31	101.36	99.35	0.61	65.67	0.048 *
1000 m	184.50 ± 13.70	186.38 ± 14.48	183.5 ± 14.12	101.00	99.52	1.65	22.47	0.005 *
1500 m	183.31 ± 15.17	188.56 ± 15.24	185.31 ± 14.23	102.90	101.27	0.84	26.22	0.041 *
2000 m	183.00 ± 14.71	186.95 ± 16.21	185.56 ± 15.12	102.14	101.53	4.92	20.06	0.001 *
2500 m	181.43 ± 15.37	187.69 ± 14.87	186.56 ± 14.92	103.50	102.92	1.69	19.01	0.006 *
3000 m	181.93 ± 14.57	186.00 ± 14.91	185.43 ± 14.46	102.24	102.03	0.67	11.50	0.122
3500 m	181.68 ± 15.08	188.50 ± 15.72	187.75 ± 15.28	103.76	103.45	0.48	19.01	0.156
4000 m	181.37 ± 16.07	188.63 ± 15.92	187.43 ± 15.42	104.05	103.47	456.28	11.11	0.001 *
4500 m	184.12 ± 15.87	190.69 ± 15.62	188.75 ± 15.96	103.61	102.60	1.23	21.03	0.016 *
5000 m	191.12 ± 17.25	199.88 ± 19.49	195.12 ± 17.37	104.60	102.22	3.30	41.32	0.001 *

Note. * indicates that a significant difference was found between tDCS and sham, *p* < 0.05.

## Data Availability

Data will be available upon request from the corresponding author.

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
