# Peer review of "Effect of Transcranial Direct Current Stimulation on Endurance Performance in Elite Female Rowers: A Pilot, Single-Blinded Study"

_brainsci, 2022, doi:10.3390/brainsci12050541_

Round 1

Reviewer 1 Report

In their study, "Effect of transcranial direct current stimulation on endurance performance in elite female rowers: a pilot, single-blinded  study",  the authors completed a pilot single-blinded study on 8 participants to assess the effects of a single tDCS session on endurance. The author hypothesized that active tDCS could enhance endurance performance during rowing. As a result, the authors report that the one-session tDCS did not successfully induce significant benefits in rowing endurance performance. 

This study provided additional evidence for the effects of tDCS on endurance. The study design is generally sound, but some of the analysis procedures are confusing. My main concerns are methodological (in relation to the choice of stimulation montage, a lack of information on what has been stimulated, and possible confounds during tDCS. For those reasons, I cannot recommend accepting the manuscript. Some comments are described below:

1. Introduction, "In addition to the peripheral elements (i.e., outputs of neural drive and spinal motoneurons excitability), multiple central elements are pertaining to endurance capacity, including the activation of cortical regions within the brain "  :  This sound too unclear,  I would suggest being more specific recent publication found that endurance can be modulated targeting primary motor cortical regions mono-laterally and also bilaterally.

2. The rationale behind cortical excitability modulation and endurance capacity is not well explained. In several studies, tDCS has been combined with TMS to investigate the modification of cortical excitability by measuring MEPs (see Nitsche & Paulus, 2000 ). In the current paradigms, cortical excitability is measured by TMS application on the left motor cortical representation of the first dorsal interosseous (FDI) to abductor pollices brevis (ABP) before and after administration anodal or cathodal tDCS.  Generally, if the anode is placed above the motor cortex, single pulse TMS (sTMS) will result in a larger motor evoked potential (MEP) after DC stimulation. MEP size will be reduced if the cathode is placed at the motor cortex. The authors argue that "Anodal stimulator was placed over M1(Cz according to the 10-20 international EEG system), while the bilateral cathodal stimulators were placed over C5, C6 " : This is not correct instead, it is likely that cathodal stimulation have been applied to both left and right M1. Using anode on Cz and two cathods C5-C6 , could stimulate not only M1  but also the supplementary motor area,  primary and secondary somatosensory cortices (S1 and S2). As shown in Figure 1.

3.   Introduction, "Recent studies have emerged to explore the effects of tDCS on the endurance performance in sports. Angius et al. [9] showed in a 50 study, for example, that one session of 20-minute tDCS targeting the primary motor cortex 51 (M1) significantly improves endurance performance in riders. Liu et al. [10] observed that tDCS targeting M1 significantly improved rowing performance on the ergometer in college rowers. ": Two papers you cited provide such evidence adopting a specific anodal bilateral tDCS montage (Angius et al) and anode stimulation over the left M1 and the cathode on the contralateral shoulder (Liu et al). Therefore, the rationale behind the montage selection is not clear and not supported by the cited evidence (Angius et al, Liu et al.) and thus cannot be used to justify the hypothesis. 

4. Figure2 is very hard to read

5. Proofreading is strongly recommended; preferably, ask for help from a native speaker.

Author Response

Thanks for considering our work. We really appreciate these critical and useful suggestions. We believe that these will significantly improve our manuscript.

Comments and Suggestions for Authors

In their study, "Effect of transcranial direct current stimulation on endurance performance in elite female rowers: a pilot, single-blinded study",  the authors completed a pilot single-blinded study on 8 participants to assess the effects of a single tDCS session on endurance. The author hypothesized that active tDCS could enhance endurance performance during rowing. As a result, the authors report that the one-session tDCS did not successfully induce significant benefits in rowing endurance performance. 

This study provided additional evidence for the effects of tDCS on endurance. The study design is generally sound, but some of the analysis procedures are confusing. My main concerns are methodological (in relation to the choice of stimulation montage, a lack of information on what has been stimulated, and possible confounds during tDCS. For those reasons, I cannot recommend accepting the manuscript. Some comments are described below:

  1. Introduction, "In addition to the peripheral elements (i.e., outputs of neural drive and spinal motoneurons excitability), multiple central elements are pertaining to endurance capacity, including the activation of cortical regions within the brain "  :  This sound too unclear,  I would suggest being more specific recent publication found that endurance can be modulated targeting primary motor cortical regions mono-laterally and also bilaterally.

Answer:

We have made revisions in the manuscript. This part of the introduction has been changed from "In addition to the peripheral elements (i.e., outputs of neural drive and spinal motoneurons excitability), multiple central elements are pertaining to endurance capacity, including the activation of cortical regions within the brain …" to “Endurance capacity is a relatively complex basic ability; it consists of the peripheral ele-ments and multiple central elements [4]. Recent studies have observed that central ele-ments, including the cortical regions, of the brain play an important regulatory role in endurance performance [5], and have linked greater cerebral excitability in brain regions pertaining to cognitive–motor control to an increased output of neural drive and spinal motor neuron excitability, effectively alleviating fatigue and increasing endurance during exercise [6-8]. Therefore, strategies designed to facilitate the excitability of cortical regions within the brain hold great promise in enhancing endurance capacity, which will ulti-mately help improve rowing performance.  Please check the specific responses from line 37 to line 46.

  1. The rationale behind cortical excitability modulation and endurance capacity is not well explained. In several studies, tDCS has been combined with TMS to investigate the modification of cortical excitability by measuring MEPs (see Nitsche & Paulus, 2000 ). In the current paradigms, cortical excitability is measured by TMS application on the left motor cortical representation of the first dorsal interosseous (FDI) to abductor pollices brevis (ABP) before and after administration anodal or cathodal tDCS.  Generally, if the anode is placed above the motor cortex, single pulse TMS (sTMS) will result in a larger motor evoked potential (MEP) after DC stimulation. MEP size will be reduced if the cathode is placed at the motor cortex. The authors argue that "Anodal stimulator was placed over M1(Cz according to the 10-20 international EEG system), while the bilateral cathodal stimulators were placed over C5, C6 " : This is not correct instead, it is likely that cathodal stimulation have been applied to both left and right M1. Using anode on Cz and two cathods C5-C6 , could stimulate not only M1  but also the supplementary motor area,  primary and secondary somatosensory cortices (S1 and S2). As shown in Figure 1.

Answer:                                                                                                                      We agree that using large sponge electrodes in Halo system may induce very varied current flow influencing other brain regions, especially using the montage here as Reviewer commented We have thus made significant changes in “Transcranial direct current stimulation (tDCS) procedure” Section at Method to make it clear.

3.Introduction, "Recent studies have emerged to explore the effects of tDCS on the endurance performance in sports. Angius et al. [9] showed in a 50 study, for example, that one session of 20-minute tDCS targeting the primary motor cortex 51 (M1) significantly improves endurance performance in riders. Liu et al. [10] observed that tDCS targeting M1 significantly improved rowing performance on the ergometer in college rowers. ": Two papers you cited provide such evidence adopting a specific anodal bilateral tDCS montage (Angius et al) and anode stimulation over the left M1 and the cathode on the contralateral shoulder (Liu et al). Therefore, the rationale behind the montage selection is not clear and not supported by the cited evidence (Angius et al, Liu et al.) and thus cannot be used to justify the hypothesis. 

Answer: Thanks. We agree! The two references are not well appropriate to support our study design. Then, we have made the revisions in this part to use the manner example a reference to support this. In part of the study, “ tDCS is one such technique that can non-invasively and safely modulate cortical ac-tivities by sending direct micro-level currents to the targeted regions via scalp electrodes [9,10]. In recent decades, more attention has been paid to the ergogenic effects of tDCS on endurance performance [5] [11]. Vieira et al. [12] and Fores et al. [13], for example, showed that using transcranial direct current stimulation (tDCS) to modulate the cortical excitabil-ity of the prefrontal cortex can significantly improve the endurance performance of ath-letes in back squat and swimming. In another study, Wang et al. [8] reported that bilater-ally modulating the cortical excitability of the primary motor cortex (M1) using tDCS can significantly improve muscle endurance performance in elbow flexion tasks.

By using Halo Sport system of tDCS, Park et al. [14] showed that one session of tDCS targeting M1 significantly improved endurance running performance at 80% VO2max in trained runners. This finding suggests that using tDCS to target M1 via Halo system can also improve the endurance performance. Still, the effects of such tDCS protocol on en-durance in elite rowing athletes have not been well characterized. Please check it from line 47 to line 60.

  1. Figure2 is very hard to read

Answer: Thanks. We have revised it, please check it from line 137 to line 138.

  1. Proofreading is strongly recommended; preferably, ask for help from a native speaker.

Answer: Thanks for your careful comments. We have completed to the English editing from MDPI for this manuscript. And we have uploaded a new manuscript into system. Please check it.

Reviewer 2 Report

The overall purpose of the current pilot study was to examine the influence of transcranial direct current stimulation (tDCS) of primary motor cortex on rowing performance in a total of 8 rowers. A single-blind, randomized, counterbalanced, SHAM controlled design was employed. All subjects completed a baseline testing session and then two counterbalanced intervention experiments (tDCS, SHAM). tDCS was applied for 20 minutes and before the rowing performance tests of interest were completed.  The main findings appeared to be somewhat mixed as tDCS did improve endurance by 1.05% vs 0.24% for SHAM, but in different parts of the paper the authors seemed to interpret the results differently. Sometimes the results were implied to be viewed as successful whereas other times they were not (see below).

The study had several strengths.

  1. Other than it being single-blinded, it was well-controlled and the design was good.
  2. The use of elite athletes and the fact that 8 were able to be recruited was the biggest strength. I think the use of a commercially available device was also a strength.
  3. The data presentation was very good.

The weaknesses were

  1. Single blind design, but this is somewhat understandable in this instance and by itself is not a fatal flaw.
  • There are numerous English problems which is understandable as it is likely the authors second language and they did infinitely better than I could have ever done in a second language. Nonetheless, it needs extensive English changes to be publishable this journal. There are many instances of incorrect grammar or awkward wording. There are too many to mention but one example is line 74 “participants were executed”. There are also some typos. One example is line 63 “hight”. Extensive proofreading by an English speaker can fix these issues.

The main issue with the paper is that as mentioned above the wording used by the authors in the abstract, results, and discussion sometimes sounds like the results are viewed as successful and other times they are not. Maybe the English had something to do with this. The authors should fix and reword these portions and make their interpretations clear.

In summary, this paper is good overall and the elite population makes it publishable eventually. However, the authors have to make the changes above before it can be published at the current time. I applaud the authors for being able to conduct it on elite performers.

Author Response

Thanks for considering our work fits into an interesting topic. We really appreciate these very helpful suggestions which we believe significantly improved our manuscript. We have made revisions in the manuscript and please check the specific responses as following.

Comments and Suggestions for Authors

This manuscript describes a study on influence of tDCS on rowing performance. The main finding was that anodal tDCS over the motor area marginally improved endurance with some limited parameters.

1) This study deals with neuroenhancement using the non-invasive brain stimulation (NIBS) technique, which can potentially lead to a sort of “doping” . Ethical issues related to this idea should be explicitly discussed.

Answer: Thanks, we agree! While tDCS is showing increasing interest in elite athletes, whether the doping effect of tDCS on sports performance in elite athletes is still unclear (Reardon, 2016, Bowman-Smart, 2020). This study has obtained support from the institutional review board committee before investigating the effect of tDCS on rowing performance in elite rowers. Meanwhile, we also recognize that there are possibly the doping effects of tDCS. Therefore, we design this pilot study to obtain data and to further investigate the doping effect size of tDCS on sports performance in elite rowers.

2) Training strategy adopted here is not clear enough. Please explain “training plan from coach” and verbal encouragement (both page 3) more concretely.

Answer: Thanks for your suggestion. We take cooperation with coaches from the rowing team, involving the performance coach and the strength and conditioning coach (S&C Coach), before starting the experimental test of tDCS. The warm-up training plan is responsible by the S&C Coach. The verbal encouragement is responsible by the performance coach. We have corrected these at the method part in our manuscript, please check line 90, line 116, and line 117.

3) Please discuss potential effect size of reducing TC500m and PC500 at limited points.

Answer: Thanks for your suggestion. We have changed the words to “Linear mixed effects models revealed significant differences in the percent changes of TC500 between tDCS and sham at 2500m (t=-4.50, p=0.011, ES=0.91) and 4000m (t=3.11, p=0.036, ES=0.84), that is, after receiving tDCS, TC500 was significantly shorter than that after sham (Figure 3 and Table 2). Significant difference in the percent changes in PC500 was also observed af-ter tDCS and that after sham at 500m (t=65.67, p=0.048, ES=0.99), 1000m (t=22.47, p=0.005, ES=0.99),1500m (t=26.22, p=0.041, ES=0.99), 2000m (t=20.06, p=0.001, ES=0.99), 2500m (t=19.01, p=0.006, ES=0.99), 4000m (t=11.11, p=0.001, ES=0.98), 4500m (t=21.03, p=0.016, ES=0.99) and 5000m (t=41.32, p=0.001, ES=0.99).”

4) “No significant difference was observed in endurance performance between two intervention visits where the interventions were operated by counterbalanced order (t=-0.421, p=0.685), lines 125--127.” What was the dependent variable for “endurance performance?” Given that a single t-value is reported, there must be only one parameter here. If I understand it correctly, a few parameters for endurance were measured, including time, 500m/split etc. Please specify.

Answer: Thanks. It is a typo. We have changed it from “endurance performance” to “endurance time”, please check line 130. 

5) Given Figure 1, both leg and hand/arm motor areas seemed to be stimulated. Is it possible to infer which could contribute more?

Answer: Thanks for this interesting question. In this sponge type electrode, the current is diffused and the modeling of Halo does not allow the estimation of regional electrical intensity. We have put this limitation at the limitation in our manuscript, please check it from line 228 to line 230.

6) The study participants were all female. Are the results also applicable to male? Why were only female participants recruited?

Answer: Thanks. This is a important bias and we agree the effects of tDCS may be different between sexes, which was shown in Nitsche et al, 2008. This is also an issue due to the reality that when we were recruiting participants, only women rowers were eligible and were with matched rowing performance. The men athletes were not at the same skill level. So to increase the homogeneity in rowing skill, we recruited only women athletes.

7) Were the participants able to discriminate real from sham tDCS? This point is important to estimate how placebo effects influenced the results.

Answer: We agree blinding is important and this is a limitation in this study as we did not explicitly examine the blinding efficacy. Only verbal report was provided by the athletes on their feelings about the stimulation but did not record the blinding. We have included it into the limitation part of discussion.

Below are some minor points:

8) 9% NaCl (line 89) would be typically 0.9%. Was it 9% with this device?

Answer: Thanks, yes, the NaCl concentration is 0.9%. The information has been corrected. Please check the content at line 94.

9) Some English errors are found, which include but are not limited to the following:

“which has shown been effectively confirmed (line 91)” should be “which has been shown effectively confirmed.”

Answer: Thanks for checking and finding these typos. The manuscript has been thoroughly revised and edited by a native speaker.

10) Letters in the figures are too small (Fig 2, 3).

Answer: Thanks for your careful comments. We have revised the font size in Fig 2 and Fig 3.

Reviewer 3 Report

This manuscript describes a study on influence of tDCS on rowing performance. The main finding was that anodal tDCS over the motor area marginally improved endurance with some limited parameters.

1) This study deals with neuroenhancement using the non-invasive brain stimulation (NIBS) technique, which can potentially lead to a sort of “doping.” Ethical issues related to this idea should be explicitly discussed.

2) Training strategy adopted here is not clear enough. Please explain “training plan from coach” and verbal encouragement (both page 3) more concretely.

3) Please discuss potential effect size of reducing TC500m and PC500 at limited points.

4) “No significant difference was observed in endurance performance between two intervention visits where the interventions were operated by counterbalanced order (t=-0.421, p=0.685), lines 125--127.” What was the dependent variable for “endurance performance?” Given that a single t-value is reported, there must be only one parameter here. If I understand it correctly, a few parameters for endurance were measured, including time, 500m/split etc. Please specify.

5) Given Figure 1, both leg and hand/arm motor areas seemed to be stimulated. Is it possible to infer which could contribute more?

6) The study participants were all female. Are the results also applicable to male? Why were only female participants recruited?

7) Were the participants able to discriminate real from sham tDCS? This point is important to estimate how placebo effects influenced the results.

Below are some minor points:

8) 9% NaCl (line 89) would be typically 0.9%. Was it 9% with this device?

9) Some English errors are found, which include but are not limited to the following:

“which has shown been effectively confirmed (line 91)” should be “which has been shown effectively confirmed.”

10) Letters in the figures are too small (Fig 2, 3).

Author Response

Thanks for considering our work fits into an interesting topic. We really appreciate these very helpful suggestions which we believe significantly improved our manuscript. We have made revisions in the manuscript and please check the specific responses as following.

Comments and Suggestions for Authors

The overall purpose of the current pilot study was to examine the influence of transcranial direct current stimulation (tDCS) of primary motor cortex on rowing performance in a total of 8 rowers. A single-blind, randomized, counterbalanced, SHAM controlled design was employed. All subjects completed a baseline testing session and then two counterbalanced intervention experiments (tDCS, SHAM). tDCS was applied for 20 minutes and before the rowing performance tests of interest were completed.  The main findings appeared to be somewhat mixed as tDCS did improve endurance by 1.05% vs 0.24% for SHAM, but in different parts of the paper the authors seemed to interpret the results differently. Sometimes the results were implied to be viewed as successful whereas other times they were not (see below).

The study had several strengths.

  1. Other than it being single-blinded, it was well-controlled and the design was good.
  2. The use of elite athletes and the fact that 8 were able to be recruited was the biggest strength. I think the use of a commercially available device was also a strength.
  3. The data presentation was very good.

The weaknesses were

  1. Single blind design, but this is somewhat understandable in this instance and by itself is not a fatal flaw.
  • There are numerous English problems which is understandable as it is likely the authors second language and they did infinitely better than I could have ever done in a second language. Nonetheless, it needs extensive English changes to be publishable this journal. There are many instances of incorrect grammar or awkward wording. There are too many to mention but one example is line 74 “participants were executed”. There are also some typos. One example is line 63 “hight”. Extensive proofreading by an English speaker can fix these issues.

The main issue with the paper is that as mentioned above the wording used by the authors in the abstract, results, and discussion sometimes sounds like the results are viewed as successful and other times they are not. Maybe the English had something to do with this. The authors should fix and reword these portions and make their interpretations clear.

In summary, this paper is good overall and the elite population makes it publishable eventually. However, the authors have to make the changes above before it can be published at the current time. I applaud the authors for being able to conduct it on elite performers.

Answer: We are very sorry for the mistakes in this manuscript and inconvenience they caused in your reading. The manuscript has been thoroughly revised and edited by a native speaker from MDPI, so we hope it can meet the journal’s standard. Thanks so much for your useful comments.
